# Understanding the Polyamine and mTOR Pathway Interaction in Breast Cancer Cell Growth

**DOI:** 10.3390/medsci10030051

**Published:** 2022-09-10

**Authors:** Oluwaseun Akinyele, Heather M. Wallace

**Affiliations:** 1Institute of Medical Sciences, School of Medicine, Medical Sciences and Nutrition, University of Aberdeen, Aberdeen AB25 2ZD, UK or; 2Division of Genetics and Genomic Medicine, Department of Pediatrics, University of Pittsburgh School of Medicine, Pittsburgh, PA 15260, USA

**Keywords:** growth, polyamines, mTOR pathway, phosphorylation, 4E-BP1, p70SK1 and translation initiation

## Abstract

The polyamines putrescine, spermidine and spermine are nutrient-like polycationic molecules involved in metabolic processes and signaling pathways linked to cell growth and cancer. One important pathway is the PI3K/Akt pathway where studies have shown that polyamines mediate downstream growth effects. Downstream of PI3K/Akt is the mTOR signaling pathway, a nutrient-sensing pathway that regulate translation initiation through 4EBP1 and p70S6K phosphorylation and, along with the PI3K/Akt, is frequently dysregulated in breast cancer. In this study, we investigated the effect of intracellular polyamine modulation on mTORC1 downstream protein and general translation state in two breast cancer cell lines, MCF-7 and MDA-MB-231. The effect of mTORC1 pathway inhibition on the growth and intracellular polyamines was also measured. Results showed that polyamine modulation alters 4EBP1 and p70S6K phosphorylation and translation initiation in the breast cancer cells. mTOR siRNA gene knockdown also inhibited cell growth and decreased putrescine and spermidine content. Co-treatment of inhibitors of polyamine biosynthesis and mTORC1 pathway induced greater cytotoxicity and translation inhibition in the breast cancer cells. Taken together, these data suggest that polyamines promote cell growth in part through interaction with mTOR pathway. Similarly intracellular polyamine content appears to be linked to mTOR pathway regulation. Finally, dual inhibition of polyamine and mTOR pathways may provide therapeutic benefits in some breast cancers.

## 1. Introduction

Breast cancer continues to be a major health challenge due to the prevalence and mortality associated with the disease. In the UK and USA, 1 in 8 women are estimated to be diagnosed of breast cancer in their lifetime, with over 11,000 breast cancer related death per year. Although, there has been a significant improvement in breast cancer management in the last decades which has led to increase in survival rate, however, some breast cancers still show resistance to current systemic therapies. This is due, in part, to multiple metabolic pathways that are activated in response to specific drug treatment and contribute significantly to cancer cells proliferation and treatment resistance [1,2,3,4].

The polyamines putrescine, spermidine and spermine are absolute requirement for normal eukaryotic cell growth but are also implicated in carcinogenesis [5,6,7,8]. Upregulation of polyamine metabolic pathway is linked to various cancers including cancers of the breast [9,10,11]. The activities of many of the enzymes involve in polyamine biosynthesis, as well as elevated polyamine content are associated with poor prognosis in breast and colorectal cancers [12,13,14], highlighting the importance of polyamines in breast carcinogenesis and disease progression. Although targeting polyamine metabolism using pharmacological agents has not been successful as a treatment strategy for breast cancer. However, combination treatments that limit the polyamine content of cancer cells have shown to sensitize both ER+(eostrogen receptor positive) and ER-negative breast cancer cells to anti-cancer agents [15,16], an indication that targeting polyamine metabolism may be therapeutically beneficial in some breast cancers when combined with other anti-cancer agents.

The mammalian target of rapamycin (mTOR) plays a central role in regulating metabolic pathways that promote cell growth and survival [17,18]. mTOR forms two distinct complexes named: mTOR complex 1 (mTORC1) and 2 (mTORC2) [19,20]. The central role of mTOR in cell proliferation is attributed to mTORC1 [21,22] and upon activation, it initiates downstream processes such as protein synthesis, ribosome biogenesis, nutrient metabolism, and cell cycle progression. These occur through phosphorylation that in/activates its downstream proteins including eukaryotic translation initiation factor binding protein, 4EBP1, and the ribosomal protein kinase 1, p70S6K1 [20,21,23]. 4EBP1 regulates translation initiation through interaction with eIF4E (translation initiation factor 4E), a component of the cap-binding translation initiation machinery [21]. The translation of ornithine decarboxylase (ODC), the enzyme that catalyzes the committed step in polyamines biosynthesis, is reported to be by a cap-dependent mechanism involving eIF4E [24,25]. The phosphorylation and activation of p70S6K1 also helps in ribosome biosynthesis, protein synthesis and in general cell proliferation [23,26].

Given that mTOR pathway regulates metabolic processes that promote cancer cell proliferation and that polyamines are needed in several of these processes, the intriguing possibility of a positive interaction between polyamines and mTOR pathway, in promoting cancer cell growth, arises and was investigated in this study. The effect of combination of a polyamine biosynthetic inhibitor and mTOR pathway inhibitor in the breast cancer cell was also assessed.

## 2. Materials and Methods

### 2.1. Materials

The following were purchased from Sigma-Aldrich (Dorset, UK): Dulbecco’s Modified Eagle Medium (DMEM), Penicillin/Streptomycin, MEM non-essential amino acids 100×, 0.25% Trypsin-EDTA, L-glutamine, putrescine, spermidine, spermine. Fetal bovine serum (FBS) was from Gibco by Life Technologies (Sao Paolo, Brazil), mTOR siRNA and negative control were from Fisher Chemical (Loughborough, UK), protease and phosphatase inhibitors cocktail from Roche (Indianapolis, IN, USA), BCA protein assay kit from BioVision (Milpitas, CA, USA). RIPA lysis buffer 10× from Millipore (Burlinton, MA, USA). The following antibodies were purchased from Abcam Biotechnology (Cambridge, UK) and used at the indicated dilutions: rabbit polyclonal anti-actin antibody (ab119716, 1:7, 500 dilution), goat polyclonal anti-rabbit IgG-HRP (ab205718, 1:10,000 dilution). Additionally, mouse monoclonal anti-ODC antibody (sc-398116, 1:100 dilution), mouse monoclonal anti-phospho-4EBP1 (sc-293124, 1:200 dilution), mouse monoclonal anti-phospo-p70S6K (sc-8416, 1:300 dilution), mouse monoclonal anti- eIF4E (sc-271480, 1:300 dilution) and goat anti-mouse IgG-HRP (sc-2005, 1:2,500 dilution) antibodies were from Santa-Cruz Biotechnology (Heidelberg, Germany). Rapamycin and the class 1 dual inhibitors of PI3K and mTORC1, NVP-BEZ235, were from Cayman Chemicals (Ann Arbor, MI, USA).

### 2.2. Methods

#### 2.2.1. Cell Culture

The human breast cancer cell lines MCF-7 and MDA-MB-231 (from ECACC), which are Eostrogen receptor positive and negative (ER+, and ER-), respectively, were used in this study. The cells were grown under standard conditions of 37 °C and 5% CO_2_ in T75-cm^2^ flasks containing 15 mL DMEM supplemented with 10% (*v*/*v*) FBS, 1% (*v*/*v*) penicillin/streptomycin and 1% (*v*/*v*) L-glutamine (for MCF-7 medium was further supplemented with 1% (*v*/*v*) non-essential amino acids). For experiments where exogenous polyamines were added to the cells, 1 mM aminoguanidine was also added to the cell culture media.

#### 2.2.2. Growth Determination

Cells were grown in 6 cm plates in duplicate and harvested at 48 and 72 h. At the indicated time point, cells were harvested by trypsin-EDTA method, transferred to 15 mL Falcon tube, and centrifuged for 5 min at 250× *g*. The supernatant was discarded, and the cell pellet resuspended in 1 mL of complete growth medium. Thereafter, 100 µL of this suspension was sampled for cell counting by trypan-blue exclusion assay. The remaining 900 µL was centrifuged and the supernatant discarded. The cell pellet was washed twice in 2 mL PBS after which it was subjected to polyamines and protein extraction.

#### 2.2.3. Polyamines and Protein Extraction

Polyamines were extracted by resuspending cell pellets in 1 mL of 0.2 M PCA (perchloric acid). Tubes were placed on ice for at least 30 min to allow extraction of acid soluble components and for protein precipitation. After incubation on ice, tubes were centrifuged for 5 min and acid soluble fraction (containing polyamines) was removed to 1.5 mL Eppendorf tube, stored at −20 °C until analyzed. The remaining precipitate was resuspended in 1 mL 0.3 M NaOH, left overnight at 37 °C and used for total protein quantification.

#### 2.2.4. MTT Assay

Cells were seeded in 96-well plates and allowed to grow for 48 h after which medium was replaced with that containing increasing concentrations of the drugs or their combinations (100 mL final volume). For combination studies, the drugs were added simultaneously. The cytotoxicity effect of the drugs on the cells was determined using the 3-(4,5 dimethylthiazol-2-yl)-2,5-diphenyltetrazolium bromide (MTT) assay. Ten microliters of MTT solution (dissolved in phosphate-buffered saline, PBS, 5 mg/mL) was added to each well and the plate incubated for a further 3 h at 37 °C. After incubation, culture media was removed and replaced with 100 µL of DMSO to solubilize the formazan crystals formed from the added MTT solution. The optical density of each well was determined on a microplate reader (Tecan Sunrise™, Mannedorf, Switzerland). Viable cell was estimated as a percentage relative to the untreated cells.

#### 2.2.5. Total Protein Quantification

Total protein was quantified by a modification of Lowry et al. [27] method. A bovine serum albumin (BSA) standard curve was prepared in the range 0 to 250 µg/mL in 0.3 M NaOH. Both samples and standard were plated in 96 well plate and exposed to a basic solution containing Cu^++^ for 10 min prior to addition of 0.13 M Folin-Ciocalteau reagent and incubated in the dark for 30 min. The plate was thereafter read on a microplate reader (Tecan Sunrise^TM^, Mannedorf, Switzerland). Total protein was estimated from the standard curve and express as mg/culture.

#### 2.2.6. Polyamine Quantification

Polyamines were quantified by LC-MS/MS (liquid chromatography–mass spectrometry) as previously described [28]. Briefly, 200 µL of samples and polyamines standard were placed in glass tubes, 10 µL of 100 µM of 1,7-diaminoheptane was added to each tube as internal standard, plus 50 µL of 1 g/mL sodium carbonate and 500 µL of freshly prepared 10 mg/mL dansyl chloride in acetone. Tubes were left overnight at 37 °C. Next day, 0.5 mL of toluene was added to extract the dansylated products. Organic phase was evaporated to dryness under N_2_ flow, the aqueous phase was reconstituted in 200 µL methanol and vortex mixed for 10 sec. This was then analyzed with LC-MS. Results were expressed in nmol and normalized to the total protein content for each sample.

#### 2.2.7. Western Blot

Cells were seeded in 10 cm dishes and exposed to drugs depending on the experiment. After drug exposure, cells were harvested and lysed on ice with ice-cold RIPA lysis buffer containing protease and phosphatase inhibitors cocktail. The cell lysates were centrifuged at 17,000× *g* for 15 min at 4 °C. The protein content of the supernatant was quantified by BCA protein assay. Samples containing equal amounts of proteins were separated on SDS-PAGE (12% gel), transferred to a nitrocellulose membrane overnight at 25 V. Membranes were blocked in 5% (*w*/*v*) fat free-milk in TBS-Tween-20 (0.1% (*v*/*v*) for 1 h on a shaker, and then incubated with primary antibodies for 3 h at room temperature on a bench roller. After washing three times for 5 min in TBST, membranes were incubated with secondary antibodies conjugated HRP for 1 h at room temperature. After incubation, membranes were washed three times for 5 min in TBST, followed by development with the enhance chemiluminescence substrate (SuperSignal West Dura, Life Technologies, Carlsbad, CA, USA). Blot images were detected using Gel Doc Imager Range (VWR, Leuven, Belgium). The images were quantified by measuring the intensity of correctly sized bands using the freely available ImageJ software and normalized to the actin band intensity for all experiments.

#### 2.2.8. Polysome Profile Analysis

Cells were seeded in T175 flasks and treated with respective drugs based on the experiment. After drug exposure, cells were briefly treated with 100 µg/mL cycloheximide for 15 min, rinsed in cycloheximide containing PBS and detached by trypsinization. The cells were then collected by centrifugation into a 50 mL Falcon tubes and the pellet washed twice with ice-cold PBS containing 100 µg/mL cycloheximide and resuspended in 500 µL of complete lysis buffer [25 mM Tris-HCl (pH 7.4), 50 mM KCl, 5 mM MgCl_2_, 2 mM DTT, 100 µg/mL cyclohexamide and 200 µg/mL heparin, 1% triton-100, 1% sodium deoxycholate and protease inhibitor]. Cells were lysed by pumping the solution up and down a 25-gauge needle three times and left on ice for 20 min while vortexing intermittently. The tubes were centrifuged at 17,000× *g* for 15 min at 4 °C. Total protein in the supernatants was estimated using nanodrop at 280 nm. Equal amount of each sample corresponding to 20 OD was carefully layered into an 11 mL 15–50% (*w*/*v*) sucrose gradient containing [25 mM Tris-HCl (pH 7.4), 50 mM KCl, 5 mM MgCl_2_, 2 mM DTT, 100 µg/mL cyclohexamide and 200 µg/mL heparin] in 11.5 mL polycarbonate tubes. The sucrose gradient tubes were ultra-centrifuged at 270,900× *g* for 2 h 15 min at 4 °C. Following centrifugation, gradients were unloaded from the top by pumping 60% sucrose solution containing 0.05% bromophenol blue to the bottom of each tube. The content of each gradient layer was measured by UV absorbance using an Optical Unit UV-1 at a wavelength of 254 nm. Results were recorded graphically with a chart recorder and redrawn using a tracer. All equipment was from Pharmacia Biotech, Uppsala, Sweden.

#### 2.2.9. siRNA Gene Knockdown

siRNA gene knockdown was carried out according to the manufacturer’s instruction. Briefly, cells were seeded in a 6 well-plate and grown to 50% confluence. Then, 1.5 µL of 50 µM siRNA stock and 1.5 µL of scrambled siRNA (50 µM) were each added to separate 250 µL serum and antibiotic-free base medium in a reaction tube, this was followed by the addition of 4 µL lipofectamine transfection solution. The mixture was gently resuspended and allowed to incubate for 10 min. Briefly before transfection, medium was refreshed in each well and the entire transfection mixture was carefully added to the wells and further incubated for the indicated time. The effect of the gene knockdown was determined by Western blot analysis of downstream protein expression.

#### 2.2.10. Statistical Analysis

Statistical analysis was performed with GraphPad Prism vs7. Values are shown as mean of all replicates ± standard error of the mean (S.E.M) in which the number of independent experiments was equal or more than three. Values were analyzed by ANOVA with Dunnett’s post-test or by students’ t-test. A *p* value less than 0.05 was considered statistically significant. Where the number of independent experiments was less than three, values shown are mean ± range.

To study the effect of drug combination on cells toxicity or viability, excess over bliss (EOB) analysis was performed to determine the drug combination effects at each combination dose according to Liu et al. [29]. An EOB score > 1 is considered synergism, a score of >0 but <1 is considered independent/additive while a score ˂ 0 is considered antagonism.

## 3. Results

### 3.1. Polyamine Modulation Alters mTORC1 Downstream Proteins Phosphorylation and Translation Status

We have previously shown that modulating breast cancer cell polyamine content altered their growth responses [28], since mTORC1 pathway is crucial for cell proliferation, we therefore investigated the effect of intracellular polyamine modulation on mTORC1 pathway. The phosphorylation of 4EBP1 and p70S6K are important in determining signaling pathway mediated through mTORC1. To determine whether polyamine modulation alters the phosphorylation status of these proteins in promoting cell proliferation, protein extracts from untreated control, DFMO treated and DFMO pre-treatment plus exogenous polyamines, were fractionated by SDS-PAGE and subjected to Western blot analysis using anti-phosphorylated 4EBP1 and p70S6K antibodies. Results showed that exogenous spermidine and spermine tends to increase both 4EBP1 and p70S6K phosphorylation after 6 h, with spermine having the greatest effect (Figure 1a,b). Since the phosphorylation of 4EBP1 causes its dissociation from eIF4E, there was also a trend increase in eIF4E protein level after 6 h treatment with spermine (Figure 1b).

mTORC1 is known to regulate translation initiation through 4EBP1 phosphorylation, and since polyamine modulation tends to alter 4EBP1 and p70S6K proteins phosphorylation, therefore the effect of intracellular polyamine modulation on translation state of the cells was monitored. Polysome profiles obtained by sucrose gradient ultracentrifugation revealed a substantial inhibition of general translation state of the cells following polyamine depletion by DFMO as seen by a decrease in heavy polysome peaks with an increase in the monosome peak (Figure 2a). Addition of exogenous spermidine after 24 h caused an increase in the polysome peaks following DFMO pre-treatment (Figure 2a). Estimation of the area under curve revealed a decrease in polysome to monosome ratio (p/m), an indication of the inhibition of translation initiation following polyamine depletion. However, addition of spermidine reversed this translation inhibition resulting in an increase in p/m ratio after 24 h (Figure 2b).

### 3.2. Inhibition of mTORC1 Inhibits Cell Growth and Decreases Polyamine Content

As shown above, polyamine pathway modulation alters mTORC1 downstream protein phosphorylation and translation initiation. On the hand, mTORC1 has been shown to regulate polyamine biosynthetic enzymes through activation of S-Adenosylmethionine decarboxylase (SAMDC) [30], a key enzyme in the synthesis of higher order polyamines (spermidine and spermine). Additionally, the expression of ODC is controlled by the cap-forming initiation factor, eIF4E. Since mTORC1 regulates eIF4E abundance through 4EBP1 phosphorylation, thus, there exist an intriguing possibility of a positive feedback interaction between polyamines and mTOR pathway in regulating breast cancer cells growth. Thus, the effect of mTOR regulation on polyamine content in the breast cancer cells was also examined using transient knockdown of mTOR gene by mTOR siRNA.

First the effects of mTOR siRNA transfection on the phosphorylation of mTORC1 downstream protein, 4EBP1, in MCF-7 cells after 48 and 72 h was measured. The phosphorylation of this protein serves as a marker of mTORC1 pathway activation. Transfection of MCF-7 cells with mTOR siRNA caused 20% decrease in 4EBP1 phosphorylation after 48 h. However, at 72 h, there was over 50% reduction in the phosphorylation of 4EBP1 in the cells (Figure 3a,b), an indication of mTORC1 pathway inhibition.

Additionally, growth determination following cells transfection with mTOR siRNA showed 25% and 35% decrease in cell number after 48 and 72 h, respectively, suggesting that the transient knockdown of mTOR inhibits the breast cancer cell growth (Figure 4a). Polyamine analyses following cells treatment with mTOR siRNA also showed decreases in putrescine and spermidine content with no changes in spermine content after 48 h and 72 h (Figure 4b). These decreases in putrescine and spermidine following mTOR knockdown indicate a possible regulation of polyamine content by mTOR pathway. Similarly, exposure of the cells to the dual PI3K/mTOR inhibitor, NVP-BEZ-235, caused decreases in growth, total polyamine content and ODC protein level (Appendix A respectively), suggesting that inhibition of mTOR pathway alters polyamine metabolism hence polyamine content. Although, rapamycin treatment only resulted in growth inhibition and a decrease in ODC protein level in MCF-7 with no effects on the total polyamine content (Appendix A).

### 3.3. Polyamine Depletion Potentiates mTOR Inhibitors

The use of mTOR inhibitors as a monotherapy has shown limited clinical success in the treatment of solid tumors. Current approaches consider a combination strategy of mTOR inhibitors and other growth pathway inhibitors. In this study, the effect of dual combination of polyamine biosynthetic inhibitor and mTOR pathway inhibitor was investigated.

Co-treatment of DFMO with rapamycin induced greater cytotoxicity effect than the individual drugs alone. In MDA-MB-231 cells, combination of 5 mM DFMO with rapamycin induced an additive effect at highest rapamycin concentration used as indicated by the EOB (excess over bliss) score (Figure 5). Also, in MCF-7, the combination of DFMO with the various rapamycin concentrations yielded additive effect (Figure 5).

Similarly, co-treatment of DFMO with rapamycin for 48 h caused greater inhibition of translation in the cells as seen with decrease polysomes peaks compared to the individual drug treatment and the untreated control (Figure 6a). Similarly, the polysome to monosome ratio showed greater decrease when compared to the individual treatment alone and with the untreated control (Figure 6b).

## 4. Discussion

Research relating to polyamines has received significant attention in the last few decades due to the important roles of polyamines in cellular growth processes [31,32,33]. Functions linked to polyamines range from gene expression to apoptosis to cell cycle progression and cell proliferation [33,34,35,36]. One important area in which the role of polyamines is constantly being explored is the link to signal transduction pathways. Polyamines are known to be involved in signaling pathways that promote cell growth particularly in cancer cells [37,38,39]. Their depletion prevents the transfer of signaling information essential for cell growth from the extracellular membrane to the nucleus [38,40]. Important pathways that interact with polyamines in promoting cancer cells proliferation includes HER2-neu signaling, MAP kinase pathway, PI3K/Akt pathway and the NF-κB pathway [38,40,41]. Similarly, the polyamine spermidine, has been suggested to mediate cell growth downstream of P13K/Akt pathway [41] that functions above the mTOR signaling pathway. In the current study, we investigated how modulation of cellular polyamines alters mTORC1 pathway proteins and a process regulated by the pathway. This study also considered how dual inhibition of polyamine and mTOR pathways might enhance the inhibition of breast cancer cells growth.

Several studies have highlighted the requirements of polyamines for cell proliferation particularly in cancers [28,42,43]. This growth effects of polyamines appear to be mediated, in part, by mTORC1 downstream proteins activation as intracellular polyamine modulation tends to alter mTORC1 downstream protein phosphorylation. The phosphorylation of 4EBP1 and eIF4E was measured 6 h after polyamine addition, based on our previous data in colorectal cancer cells where intracellular polyamines shows uptake saturation after 3–4 h of cells exposure to exogenous polyamines [44]. The phosphosite Ser434 on p70S6K, assessed here, is located in the carboxyl terminal of the protein in the region described as the autoinhibitory pseudosubstrate domain, phosphorylation of this site along with other phosphosites in this region is known to prime p70S6K exposing other domains (including the Thr389, a site directly phosphorylated by mTOR) for phosphorylation, thus contributing to full activation of p70S6K1 [45,46].

The block in translation initiation as seen with the loss in polysomes that resulted in increase in monosome peak, following polyamines depletion, suggest involvement of polyamines in translation initiation, the recovery in polysomes formation by spermidine support the translation initiation regulation of polyamines. Since mTORC1 regulates translation initiation through regulating the formation of cap-binding protein complex, and polyamine modulation tends to alter the activation of mTORC1 downstream proteins phosphorylation, thus it is likely that polyamines regulate translation initiation through mTORC1 pathway activation. This, coupled with the activation of the translation elongation factor 5A, eIF5A, by spermidine [47,48,49], in part, helps to promote protein synthesis and general cell growth.

The mTORC1 pathway is a master regulator of metabolic growth processes and the polyamines are required in several of these processes for continuous cell proliferation. It appears that intracellular polyamines are linked to mTOR pathway regulation as transient knockdown of mTOR caused a decrease in polyamine content in the breast cancer cells. This further supports studies that have shown that the activation of some polyamine biosynthetic enzymes; ODC and AMD1 (S-adenosylmethionine decarboxylase) [24,30] are linked to mTOR pathway to control polyamine content, a suggestion of possible interaction between polyamines and mTOR pathway in promoting cancer cell proliferation.

One of the current approaches using mTOR inhibitors in cancer treatment involve their combination with other inhibitors that target critical metabolic pathways essential to cancer cell proliferation [50,51,52]. Additionally, the combination of DFMO with other chemotherapeutic agent have shown better therapeutic benefits in in vitro and in vivo cancer models [16]. That the combination of DFMO with rapamycin induced greater toxicity in both cell lines than individual drug alone, indicates that polyamine depletion enhances the growth inhibitory effects of rapamycin in the breast cancer cells. This enhanced effect was further confirmed by the translation analysis of the cells which was greatly inhibited when DFMO was combined with rapamycin compared to individual drugs alone. This preliminary data suggests that a combination of polyamine biosynthetic and mTOR pathway inhibitors may have greater effect in inhibiting cellular processes and pathways that promote cancer cell proliferation. Also, this may serve as potential therapeutic angle needed to be explored for treatment of breast cancer with upregulated polyamine and mTOR pathways. In the future, series of further studies that detail the cellular and molecular mechanisms of these drugs combination in the breast cancer cells will be investigated.

In conclusion, breast cancer cell growth is stringently linked to polyamine metabolism and one of the ways by which polyamines promote cell growth was shown in this study to possibly involve activation of mTORC1 pathway and induction of translation. Also, intracellular polyamine content appears to be associated with mTORC1 pathway activities. Finally, this study suggests that a combination of polyamine biosynthetic inhibitor and mTOR pathway inhibitors might induced greater therapeutic benefit in some breast cancers.

## Figures and Tables

**Figure 1 medsci-10-00051-f001:**
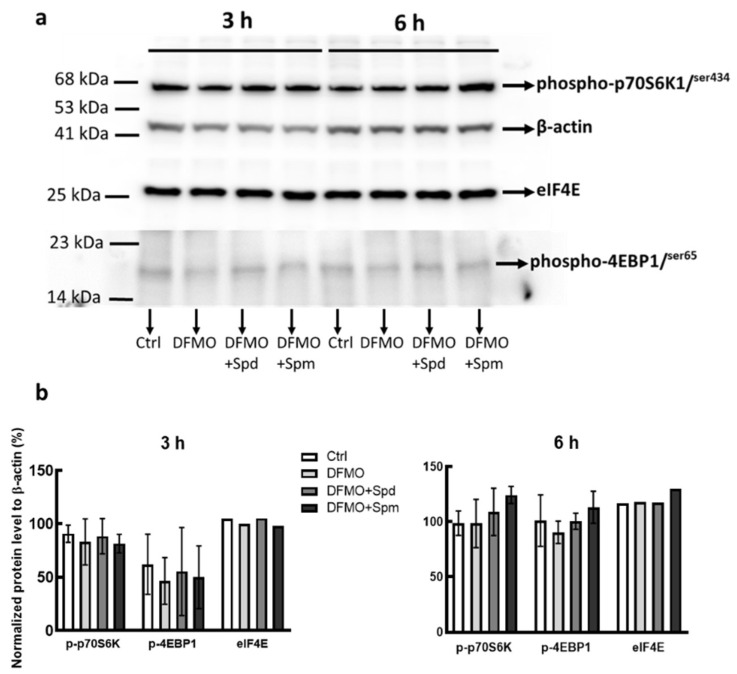
Effect of polyamine modulation of mTORC1 downstream proteins phosphorylation. MDA-MB-231 cells were grown in 5 mM DFMO containing growth medium with no amino acids supplementation for 48 h. Cells were thereafter serum starved for 12 h after which medium was refreshed and supplemented with 100 µM and 50 µM spermidine and spermine, respectively, for the indicated time. Thereafter cells were harvested and processed for Western blot protein expression determination. (**a**) Level of phosphorylated 4EBP1 and p70S6K as well as the level of eIF4E were determined on 12% SDS PAGE. β-actin was used as a loading control, Ctrl represent untreated control (**b**) Gel densitometry tracing from (**a**) was estimated using ImageJ software. Values shown are mean ± range, for two independent experiments for 4EBP1 and p70S6K.

**Figure 2 medsci-10-00051-f002:**
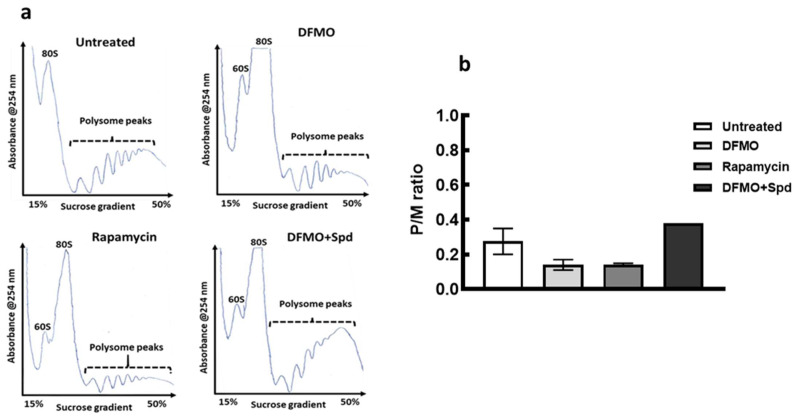
Effects of polyamine modulation on translation initiation of MCF-7 cells. MCF-7 cells were seeded, treated with DFMO for 72 h or DFMO for 48 h plus Spd for 24 h. For rapamycin, cells were treated for 72 h. For the untreated control, cells were seeded and grown for 96 h. All plates were thereafter harvested and processed for polysome analysis. (**a**) is a representative of polysome profiles from untreated control, 5 mM DFMO treatment, 0.1 μM rapamycin and 5 mM DFMO + 100 μM spermidine (Spd) for two independent experiments. (**b**) Estimation of the polysome to monosome ratio for the respective treatment for the area under curve. Values shown are mean ± range, for two independent experiments. For DFMO+Spd, the error bar is within the graph. Similar results were obtained for MDA-MB-231 cell line. Note: rapamycin treatment in this figure serves as a positive control of the effect of mTORC1 inhibition in the cancer cells.

**Figure 3 medsci-10-00051-f003:**
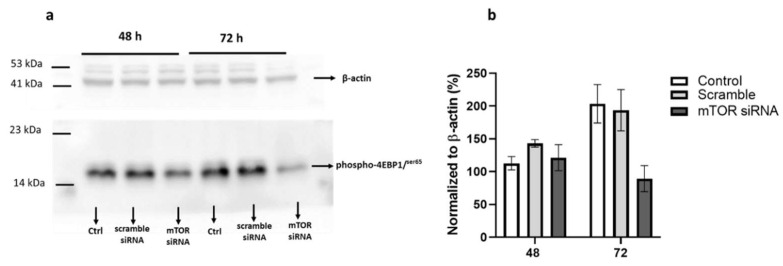
Effect of mTOR knockdown on 4EBP1 phosphorylation. MCF-7 cells were seeded and grown to 50% in 6-well plate. Cells were then transfected with mTOR siRNA (30 nM final concentration) for 48 h and 72 h, thereafter cells were harvested, (**a**) level of phosphorylated 4EBP1 was determined by SDS PAGE. β-actin was used as a loading control. (**b**) Gel densitometry tracing from (**a**) using ImageJ software. Ctrl represent untreated control. Values shown are mean ± range for two independent experiments.

**Figure 4 medsci-10-00051-f004:**
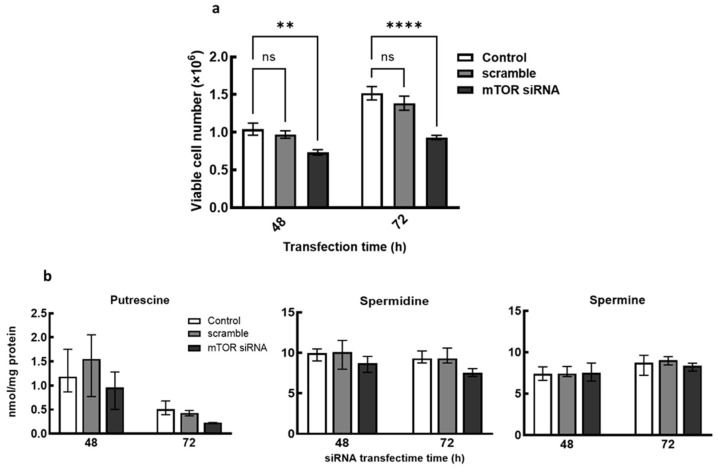
Effect of mTOR siRNA gene knockdown of growth and polyamine content. MCF-7 cells were grown to 50% confluence in 6-well plate and transfected with mTOR siRNA (30 nM final concentration) for 48 h and 72 h, thereafter cells were harvested. (**a**) Cell number was determined after 48 h and 72 h mTOR siRNA transfection by trypan-blue exclusion assay. (**b**) Polyamine content of cells after mTOR siRNA transfection was estimated by LC-MS. Values shown are mean ± S.E.M, for three independent experiments for (**a**) and mean ± range, for two independent experiment for (**b**), with two replicates per treatment and analyzed using ANOVA and Dunnett’s post-test, where ** *p* < 0.01, **** *p* < 0.0001. Control represents untreated cells, ns represents no significant difference.

**Figure 5 medsci-10-00051-f005:**
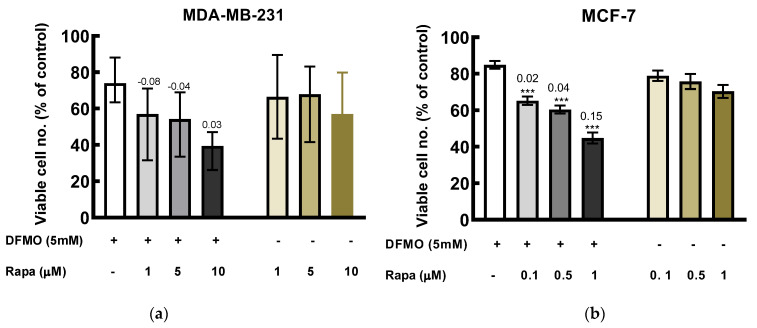
Effect of DFMO and rapamycin on viability of breast cancer cells. Cells were treated with DFMO and rapamycin, singularly and in combination at the indicated concentrations for 48 h, cytotoxicity was measured using MTT assay. (**a**) MDA-MB-231 cells in the presence of DFMO and rapamycin. (**b**) MCF-7 cells in the presence of DFMO and rapamycin (because MCF-7 cell was more sensitive to rapamycin treatment, lower concentrations was used). The results were expressed as the percentage of viable cells compared to the untreated control. Values shown are mean ± S.E.M, of at least 18 wells from three independent experiments for MCF-7 and mean ± range, of at least 12 wells from two independent experiments for MDA-MB-231. Note: Rapa represents rapamycin. Statistics was performed using one-way ANOVA with Dunnett’s post hoc analysis, where *** *p* < 0.001 vs. DFMO alone. The values above the bars for the combination treatment are the EOB scores.

**Figure 6 medsci-10-00051-f006:**
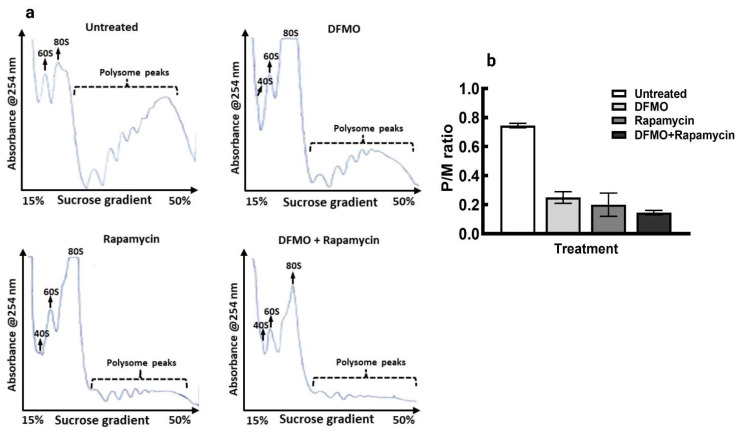
Polysome profile analysis of MCF-7 cells following treatment with DFMO and Rapamycin and the combination of both. Representative of polysome profiles of cells grown for 72 h with no treatment (**top left**), cells grown for 24 h followed by 5mM DFMO treatment for 48 h (**top right**), cells grown for 24 h followed by treatment with 0.1 µM rapamycin for 48 h (**bottom left**), and cells grown for 24 h followed by 5 mM DFMO plus 0.1 µM rapamycin treatment for 48 h (**bottom right**). Cells were harvested and processed for polysome analysis, (**a**) Images for each treatment shown is a representation for two independent experiments. (**b**) Estimation of polysome to monosome ratio from (**a**). Values shown are mean ± range for two independent experiments for all treatments.

## Data Availability

All data generated in this study are reported within the text.

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
