# Peer review of "Understanding the Polyamine and mTOR Pathway Interaction in Breast Cancer Cell Growth"

_medsci, 2022, doi:10.3390/medsci10030051_

Round 1

Reviewer 1 Report

This work by Akinyele and Wallace analyzes the effects of modulating polyamine metabolism and mTOR-dependent signaling on two breast cancer cell lines, one ER-positive and one triple negative. Both polyamine depletion with DFMO and mTORC1 inhibition with rapamycin are shown to inhibit general translation, and co-treatment of DFMO with rapamycin induced greater cytotoxicity than either drug alone, suggesting this combination may warrant further testing in other breast cancer models. There are, however several points that are not clearly explained, particularly about the interaction between the two pathways:

Points to address:

In Figure 1, it is an overstatement to say that exogenous spermidine and spermine caused an increase in both 4E-BP1 and p70S6K phosphorylation after 6 h incubation. The results cannot be analyzed for significance since they are duplicate measurements, therefore is can only be said to be a trend toward increased phosphorylation.

Can the effects of rapamycin in Figure 2 be rescued by Spd?

Are the results in Figure 2 the range for duplicate experiments? This should be stated in the figure legend.

It is a good idea to use mTOR knockdown rather than rapamycin treatment in the experiments described in Figures 3 and 4, since there are known rapamycin-independent effects, particularly of 4EBP1. However, what is the rationale for measuring phosphorylation of p70 S6-kinase Ser434? Phosphorylation of Thr389 most closely correlates with p70 kinase activity in vivo. 

What is the rationale for the high concentrations of rapamycin used in Figure 5? Treatment with 100 nM rapamycin is enough to completely inhibit mTORC1 in most systems.

MCF-7 cells are more sensitive to rapamycin plus DFMO in Figure 5, yet mTOR knockdown has little effect on MCF-7 polyamines in Figure 4 and rapamycin has no measurable effect on total polyamine content, as stated in lines 283-285 referring to Supplementary Figure 2. Similarly, DFMO has little if any effect on p70S6K and 4EBP1 phosphorylation (Figure 1). Doesn’t this suggest that, at least in these cells, mTOR and polyamine metabolism are on parallel rather than convergent pathways? 

What is the nature of the difference in effect between the two cell lines in Figure 5? Differences in mTOR activity? ODC and polyamines? Growth rate? This is not mentioned in the discussion.

It is not clear from the Figure 1 legend whether the results shown are for MCF-7 or MDA-MB-231 cells.

Author Response

Response to comments:

  1. We have changed the statement to reflect a trend towards an increase in the phosphorylation of both 4EBP1 and p70S6K following treatment of the cells with exogenous polyamines as suggested by the reviewer.
  2. We do not believe that the effect of rapamycin in figure 2 can be reversed especially as we believe that rapamycin might be acting upstream of polyamines. The effect of rapamycin in the figure was to serve as a positive control to compare the effect of polyamine depletion by DFMO and the effect of DFMO plus exogenous polyamine on the translation state of the cells to a known inhibitor of mTOR signaling pathway that also inhibits cell translation. 
  3. The results in figure 2 are a range of duplicate experiments. This has been stated in the figure legend. 
  4. Complete p70S6K activation involves phosphorylation at the Thr389 by mTORC1, however, sequential phosphorylation at proline-directed residues in the putative autoinhibitory pseudosubstrate domain, containing the ser434 phosphosite, has been suggested to contribute to p70S6k activation.
  5. We did not observe changes in the cell growth and polyamine metabolic enzymes at a lower concentration of rapamycin in these cells, hence the use of higher concentration to try to determine mechanistic effects following treatment with the mTOR inhibitor.
  6.  mTOR knockdown caused decreases in putrescine and spermidine content after 72 h with no changes in spermine (Figure 5) which is largely expected given that the polyamine biosynthetic inhibitor, DFMO has caused a slight decrease in spermine content after a similar treatment time. While rapamycin has no effect on cellular polyamine content, treatment with the dual PI3K and mTOR inhibitor caused decreases in polyamine and ODC protein levels. Similarly, rapamycin treatment caused decreases in ODC protein levels in MCF-7 cell lines (Supplementary figure 1b). Also, polyamine depletion by DFMO tends to decrease p70S6K and 4EBP1 phosphorylation (Figure 1). Taken together, these data suggest some possible interaction between the two pathways in these cells.  
  7. While we are not entirely sure what the differences in effect between the cell lines in figure 5 were, however, we believe these might be due to the differential growth rate of the cells or the class of breast cancer each cell line belongs to. MCF-7 is a hormone receptor-positive breast cancer cell line while the MDA-MB-231 cell line is a triple-negative breast cancer cell that is highly proliferative and more metastatic. 
  8. The results shown in the figure are for the MDA-MB-231 cell line and has been indicated in the legend.

Reviewer 2 Report

The role of polyamines is still poorly understood although their importance for cell health and growth is recognized. As a central controller of cell growth and translation, mTOR senses growth factor, energy and nutrient inputs. Its influence on metabolic pathways, particularly amino acid biosynthesis has an impact on polyamine biosynthesis. Conversely, polyamine levels can determine the rate of translation and thus modulate the effects of mTOR signaling. Thus the interaction between polyamines and mTOR signaling is a potentially fruitful area for investigation. The authors perform such a study on breast cancer cells in vitro, with an eye on potential therapeutic applications.

            Unfortunately the results shown are weak and in many cases lack novelty (confirm previous reports). The weak effects seen do not support the accompanying assertions and conclusions drawn. While there is promise in the initial idea, the curent manuscript is too weak and preliminary for publication. More detailed criticisms below:

 1) Fig 1The effect of polyamine depletion/repletion on mTOR activity read-outs are weak to non-existent. The mTORC1 target, p70S6K, is probed with a phospho-antibody against Ser434 (mistakenly labeled as Ser484, which is inconsistent with the antibody listed in the Materials and Methods sc-8416). The phosphosite used to measure mTORC1 activity is Thr389 which is accepted in the field and for which there are excellent antibodies. As an AGC kinase, S6K has an auto-inhibitory pseudosubstrate domain whic is phosphorylated by upstreamkinases or een auto-phosphorylated by S6K itself. Ser 434 is one such site in the presudosubstrate domain and is not a read-out for mTORC1 activity. It is thus not surprising that there is very little to no change in its phosphorylation under the various treatments despite the authors' claims in the text. Similarly any change, if any in  eIF4Elevels or phosphorylated 4EBP1 are so small as to be negligible. The authors do not perform statistical tests on the quantification but these appear to be non-significant changes.

 2) Fig 2 confirms previously reported findings that inhibition of mTORC1 by rapamycin represses general translation, and that depletion of intracellular polyamine pools by preventing its synthesis with DFMO also has the same effect. Thus there is no novelty to these findings.

 3) Fig 3 The mTORC1 activity read-out using S6K pSer434 has to be ignored for reasons given above. Depleting mTOR activity using siRNA against mTOR is a blunt tool. As they are concentrating on mTORC1 dependent effects (vis S6K and 4EBP), it would be better to specifically target mTORC1 activity with si- or sh-RNAs against the mTORC1 specific sub-unit Raptor (with controls targeting the mTORC2 specific sub-unit Rictor). Depleting mTOR itself is lethal and the effects seen may just be a consequence of cell death rather than any specific response.

 4) Fig 4 For the same reasons given in point (3), these effects could be due to cell death rather that a specific outcome.

 5) Fig 5 The effect of DFMO and rapamycin co-treament  on cell growth are only seen at 10microM of rapamycin. This is an extremely high concentration of rapamycin (in general 100nM is already more than sufficient) suggesting that the effect seen is non-specific.

 6) Fig 6 Same reason as in point (2), these results are not novel.

Author Response

Response to comments:

  1. The phospho-antibody used for p70S6K was again its Ser434 phosphosite, as pointed out by the reviewer. We apologize for this typographical error and this has been corrected in the figures. While incubation with polyamines does not cause a dramatic increase in the phosphorylation of p70S6K and 4EBP1, there is a trend increase in the phosphorylation of these proteins. While phosphorylation of p70S6K at the Thr389 is used as a readout of mTORC1 activity, sequential phosphorylation at proline-directed residues in the putative autoinhibitory pseudosubstrate domain (that contain ser434) has been suggested to contribute to p70S6k activation.
  2. We acknowledge that previous studies have shown that polyamine depletion represses global translation state of cells, however, our argument is that this process, along with other known mechanistic processes including inhibition of hypusination eIF5A, might involve decrease activation of mTOR signaling pathway. The effect of rapamycin in the figure was to serve as a positive control to compare the effect of polyamine depletion by DFMO and the effect of DFMO plus exogenous polyamine on the translation state of the cells to a known inhibitor of mTOR signaling pathway that also inhibits cell translation.
  3. While depleting complex I or II specific proteins might represent a reasonable tool to measure the effect of the mTORC1 on polyamine metabolism and on general cell growth, we however do not believe that depleting mTOR level by siRNA is a blunt tool as indicated by the reviewer as the similar effect that would be expected with any of the complexes specific protein was observed with mTOR depletion.
  4. We did not observe cell death following exposure of the cells to siRNA, what we observed was a decrease in the growth rate of the cells.
  5. At lower concentrations, we did not observe much effect on the growth of the cells, hence the use of higher concentrations. These effects are in our cells under the growth conditions and thus might be specific.
  6. To the best of our knowledge, we have not seen a publication describing the effect of the dual combination of DFMO and rapamycin on the polysome profile of a cell, thus the results reported are novel. Also, while we acknowledged that some studies have reported the polysome distribution of different cells following rapamycin or DFMO treatment, we are not sure if this has been done in the cell types used in the current study.

Round 2

Reviewer 1 Report

The authors have answered reviewer concerns adequately and the paper is now acceptable for Medical Sciences.

Reviewer 2 Report

The authors have not addressed any of the concerns raised in the original review. The main flaw is a lack of novelty (most of the results have already been described), and the effects seen are very minor. The results and overall manuscript are preliminary but even then, it would be more appropriate to shift the focus of the study to find a novel or clinically relevant aspect.